# Organization and Unconventional Integration of the Mating-Type Loci in *Morchella* Species

**DOI:** 10.3390/jof8070746

**Published:** 2022-07-19

**Authors:** Hongmei Chai, Ping Liu, Yuanhao Ma, Weimin Chen, Nan Tao, Yongchang Zhao

**Affiliations:** 1Biotechnology and Germplasm Resources Research Institute, Yunnan Academy of Agricultural Sciences, Kunming 650205, China; chm621@aliyun.com (H.C.); liuping0606@126.com (P.L.); jianxyjian@126.com (Y.M.); chwmkm@aliyun.com (W.C.); tn1953@126.com (N.T.); 2Yunnan Provincial Key Lab of Agricultural Biotechnology, Kunming 650205, China; 3Key Lab of Southwestern Crop Gene Resources and Germplasm Innovation, Ministry of Agriculture, Kunming 650205, China

**Keywords:** mating idiomorph, heterothallism, morel, alternative splicing

## Abstract

True morels (*Morchella* spp.) are a group of delicious fungi in high demand worldwide, and some species of morels have been successfully cultivated in recent years. To better understand the sexual reproductive mechanisms of these fungi, we characterized the structure of the mating-type loci from ten morel species, and seven of them were obtained using long-range PCR amplification. Among the studied species, eight were heterothallic, two were homothallic, and four types of composition were observed in the *MAT* loci. In three of the five black morel species, the *MAT1-1-1*, *MAT1-1-10*, and *MAT1-1-11* genes were in the *MAT1-1* idiomorph, and only the *MAT1-2-1* gene was in the *MAT1-2* idiomorph, while an integration event occurred in the other two species and resulted in the importation of the *MAT1-1-11* gene into the *MAT1-2* idiomorph and survival as a truncated fragment in the *MAT1-1* idiomorph. However, the *MAT1-1-11* gene was not available in the four yellow morels and one blushing morel species. *M. rufobrunnea*, a representative species of the earliest diverging branch of true morels, along with another yellow morel *Mes-15*, were confirmed to be homothallic, and the *MAT1-1-1*, *MAT1-1-10*, and *MAT1-2-1* genes were arranged in a tandem array. Therefore, we hypothesized that homothallism should be the ancestral reproductive state in *Morchella.* RT-PCR analyses revealed that four mating genes could be constitutively expressed, while the *MAT1-1-10* gene underwent alternative splicing to produce different splice variants.

## 1. Introduction

The sexual life cycle of fungi is not only involved in the production of offspring but also is important in generating variability in the populations, which may facilitate the long-term survival of fungi under unfavorable conditions [1]. In ascomycetes, sexual reproduction is usually controlled by a single mating-type locus, which contains one of two dissimilar DNA sequences occupying the same chromosomal locus [2]. These large DNA sequences were described using the word “idiomorph” to indicate their sharing little homology and not being true alleles [3]. To describe different mating genes from various species, Turgeon and Yoder [4] proposed a standardized nomenclature for idiomorphs. The *MAT1-1* idiomorph was defined as containing the *MAT1-1-1* gene, which was characterized by an alpha-box protein domain, while the *MAT1-2* idiomorph harbored the *MAT1-2-1* gene that encoded a protein with a high-mobility group (HMG) domain [4]. Besides the core mating genes *MAT1-1-1* and *MAT1-2-1*, additional genes present at the *MAT* loci are lineage-specific and conserved at the family and class levels [5]. To date, 10 additional *MAT1-1* genes (*MAT1-1-2* to *MAT1-1-11*) and 11 other *MAT1-2* genes (*MAT1-2-**2* to *MAT1-2-12*) have been identified from the *MAT* locus of various species [6,7,8].

In the heterothallic ascomycete species, individual isolates usually contain either the *MAT1-1* or the *MAT1-2* idiomorph in each haploid nucleus, and sexual reproduction requires two haploid strains containing opposite mating types. However, the primary homothallic species carrying two mating types (linked or unlinked) in the haploid genome [2] and pseudohomothallic species possessing nuclei of opposite mating types in multinuclear isolates are all capable of self-fertility and outcrossing [9].

True morels (*Morchella* spp., Morchellaceae, Pezizomycotina, Ascomycota), a group of highly-prized edible fungi with exceptional taste and nutritional value, are found in most parts of the world and are extensively traded. However, inconspicuous morpho-ecological differences make the classification of morel species confusing [10], and Genealogical Concordance Phylogenetic Species Recognition (GCPSR) based on multi-locus data (ITS, *RPB1*, *RPB2*, LSU, and *EF1**-**a*) is employed in *Morchella* systematics [11,12]. Three evolutionary clades were designated, including the basal Rufobrunnea clade (blushing morels), the Esculenta clade (yellow morels), and the Elata clade (black morels), while the phylospecies that had not been described were informally assigned the prefix codes “*Mes*” (for the Esculenta clade) and “*Mel*” (for the Elata clade), followed by a unique Arabic number [11]. To date, over 80 species-level lineages in the genus *Morchella* have been recognized worldwide, although the approaches focus more on molecular phylogenetics than morphology to identify cryptic species [13].

In *Tuber* and *Cordycepes*, some studies about the reproductive modes and mating genes have been reported [14,15]. In *Morchella*, by analyzing the mating-type ratios of single-ascospore populations, 15 species of the Elata clade and 19 species of the Esculenta clade have been confirmed to be heterothallic species [16]. The genome data of two ascospore isolates of *M. importuna* with opposite mating-type have revealed that the lengths of *MAT1-1* and *MAT1-2* idiomorphs are 10.5 and 6.7 kb, respectively [17]. On the other hand, the *MAT1-2* idiomorph harbors a single *MAT1-2-1* gene, while the *MAT1-1* idiomorph carries *MAT1-1-1* and two more newly-described mating genes, namely *MAT1-1-10* and *MAT1-1-11*, which were first reported and named from *Morchella* [7,17]. The *MAT* locus of *Morchella* sp. *Mes-**20* was verified using long-range polymerase chain reaction (PCR) amplification. The *MAT1-2* idiomorph is 7.5 kb in length, harboring a single *MAT1-2-1* gene, whereas the *MAT1-1* idiomorph is 7.8 kb in length and carries both *MAT1-1-1* and *MAT1-1-10* genes [7]. Similarly, the ancestral companions of the *MAT* locus in euascomycetes, *APN2* and *SLA2*, were located on either side of the mating idiomorph of *M. importuna* and *Mes-20*, while *SDH2* and *MBA1* joined the 5′ and 3′ end of the mating idiomorph, respectively [16].

There is a lack of knowledge on the structures and mating systems of the *MAT* locus for the other morel species. However, this information may provide better insight into the evolutionary history of *Morchella* and help identify the confusing species. In this study, we characterized the mating loci of several species from three morel clades, respectively, and sought the probable ancestral state of mating systems in this genus.

## 2. Materials and Methods

### 2.1. Species and Strains

Ten single-ascospore strains, a *MAT1-1* and a *MAT1-2* strain for each of the five morel species: *M. eximia*, *M. purpurascena*, *M. pulchella*, *Mes-6*, and *Mes-19*, were isolated in this study (Table 1), while the other strains YAASMHL1 and YAASMHL47 of *M. sextelata*, M115 of *Mes-15*, and YAASMVR of *M. rufobrunnea* were obtained from previous studies [7,17]. All strains were stored at the Mushroom Center of Yunnan Germplasm Bank of Crops at the Yunnan Academy of Agricultural Sciences, Kunming, China.

### 2.2. PCR Amplification and Sequencing of The Mating Idiomorphs

Improved primer pairs p11-1f/p11-2r, based on the p11-2f/p11-2r reported by Chai et al. [7], were used to amplify the *MAT1-1* and *MAT1-2* idiomorphs of six morel species, except for *Mes-15* and *M. rufobrunnea.* The appropriate information of *Mes-15* was derived from the genome data under accession number GWHBJCX00000000 in the Genome Warehouse in the National Genomics Data Center. For *M. rufobrunnea*, in which the primer pairs p11-1f/p11-2r were unsuccessful, the full-length sequences of the *MAT* locus were obtained using a roundabout method. First, two fragments of the following regions *SDH2* to *MAT1-1-1* (p11-2f/p1-3r) and *MAT1-2-1* to *MBA1* (p1-4f/p2-3r), were amplified. The primer pairs p1-3f/p1-3r and p1-4f/p1-4r could amplify the complete *MAT1-1-1* and *MAT1-2-1* genes, respectively, of *M. rufobrunnea* [7]. The primer p2-3r is located on *MBA1* and is used to verify the *MAT1-2* idiomorph of *M. importuna* [17]. Then, these two fragments were sequenced. According to the information of the resultant sequences, the primer pair p11-4f/p11-4r was designed to amplify the entire *MAT* locus (*SDH2* to *MBA1*) of *M. rufobrunnea*, and the corresponding long-fragment PCR products were sequenced for re-verification. The above-mentioned primers are listed in Appendix A.

The strains were maintained on potato dextrose agar medium (PDA) at 23 °C in the dark. Total DNA was isolated from 7-day-old mycelia using the Fungal gDNA Isolation Kit (BW-GD2416) (HangZhou Beiwo Meditech Co., Ltd., Hangzhou, China). All amplifications of mating idiomorphs were performed using long-range PCR, as reported by Chai et al. [7]. A 50 µL PCR reaction mixture contained 5 µL of 10 × LA PCR buffer (TaKaRa Bio, Dalian, China), 8 µL of each dNTP (2.5 mM/L), 2 µL each of forward and reverse primer (10 µM/L), 20 ng of template DNA and 0.5 U of TaKaRa LA Taq. The following cycling conditions were used: denaturation at 94 °C for 1 min, followed by 35 cycles at 98 °C for 10 s and 68 °C for 9 min, and a final extension at 72 °C for 10 min.

Once the PCR amplification was confirmed by electrophoresis, the PCR products were purified using E.Z.N.A.^®^ Gel Extraction Kit (Omega Bio-Tek, Inc., Norcross, GA, USA). The PCR products were sequenced with the PCR primers from both 3′ and 5′ termini, after which the resultant sequences were used to design new internal sequencing primers, and complete sequencing coverage in both directions was eventually achieved after several cycles. In cases where a portion of the sequences could not be sequenced smoothly for its special structure, this segment was amplified using internal primer pairs just near the special structure and the corresponding template of mating idiomorph amplicon, after which it was cloned in *Escherichia coli* DH5α using the pMD^TM^19-T vector-cloning Kit (TaKaRa Bio, Dalian, China) and sequenced in both directions using the general primers M13R-48 and M13F-47. Eventually, all sequences were assembled using the software SeqMan^TM^II 5.00 (DNASTAR Inc., Madison, WI, USA). Sequences relevant to our analyses have been deposited at GenBank (Table 2).

### 2.3. Transcriptional Analyses of The Mating Genes

Using a TaKaRa MiniBEST Plant RNA Extraction Kit, the total RNA was extracted from the mycelia that had been incubated on PDA for 10 d. Then, the first-strand cDNAs were synthesized using an Oligo dT primer (PrimeScript II 1st Strand cDNA Synthesis Kit; TaKaRa Bio, Dalian, China). The primer pairs (Appendix A) were designed to verify the coding regions of the mating genes according to the corresponding information in the mating idiomorphs, ensuring the amplification of all introns. As the design strategy utilized the conserved regions, several primer pairs could be used in some closely-related species. The PCR products were cloned into *Escherichia coli* and sequenced using the general primers M13R-48 and M13F-47. Introns were verified manually by comparing the RT-PCR sequences with the corresponding DNA sequences.

All PCR primers were designed using the software Primer Premier 5.0 (Premier Biosoft International, Palo Alto, CA, USA). The synthesis of primers and sequencing of all PCR products were performed at the Beijing Tsingke Biotechnology.

### 2.4. Comparison of The MAT Loci

To compare the organization of the *MAT* loci, the algorithm BLASTn [18] was used to assess the similarity between sequences. Based on the results of BLASTn, the analysis of synteny was performed using the software EasyFig version 2.2.5 [19].

## 3. Results

### 3.1. PCR Amplifications of The Mating Idiomorphs

The mating idiomorphs of the 12 single-ascospore strains, a *MAT1-1* and a *MAT1-2* strain for each of the four black morel species, *M. sextelata*, *M. eximia*, *M. purpurascena*, and *M. pulchella*, as well as two yellow morel species, *Mes-6* and *Mes-19*, were successfully amplified using the primer pair p11-1f/p11-2r. Primers p11-1f and p11-2r are located in the 5′ and 3′ flanking genes *SDH2* and *MBA1* of the mating idiomorph, therefore, the ends of the amplicons carried partial sequences of *SDH2* and *MBA1.* Sequence analysis indicated that the lengths of these amplified fragments varied from 8.7 to 18.4 kb. The *MAT1-2* amplicons of strains YAASMQM-4 and YAASMHL47 were shortest at 8.7 and 9.0 kb, whereas the similar *MAT1-2* fragments of the strains YAASM131-12 and YAASM23-22 were longest at 18 and 18.4 kb, while the other amplified fragments of *MAT1-1* or *MAT1-2* varied between 9.8 and 13.8 kb (Figure 1).

For strain YAASMVR from the Rufobrunnea clade, the primer pair p11-1f/p11-2r was unsuccessful in amplifying the mating locus, and several attempts were made using the possible primers reported by Chai et al. [7,17]. Eventually, two fragments involving the regions *SDH2* to *MAT1-1-1* and *MAT1-2-1* to *MBA1* were obtained and sequenced. Based on this, the primer pair p11-4f and p11-4r, which were located on *SDH2* and *MAB1*, respectively, were designed to amplify the entire mating locus. Using these primers, the lengths of the corresponding amplicons were about 17.8 kb (Figure 1).

### 3.2. Organization of The Mating Idiomorphs

A total of 18 mating idiomorphs from nine morel species, as well as a mating locus from *M. rufobrunnea*, were analyzed. Of these, the mating idiomorphs from *M. importuna* and *Mes-20* had been characterized previously [7,17], those of *Mes-15* were obtained from a draft genome database in our previous study [17], and the others were obtained in this study. The gene arrangements at the mating idiomorphs showed four compositions in these ten species (Figure 2). In the three black morel species *M. sextelata*, *M. eximia*, and *M. importuna*, the *MAT1-1* idiomorph contained three complete-mating genes, which were *MAT1-1-1*, *MAT1-1-10*, and *MAT1-1-11*, while the *MAT1-2* idiomorph contained the gene *MAT1-2-1*, which was classified into the *M. importuna* type. A second organization was observed in two black morel species, *M. purpurascena* and *M. pulchella*, and defined as the *M. purpurascena* type. This type of organization was more special, as the *MAT1-2* idiomorph carried two complete-mating genes *MAT1-2-1* and *MAT1-1-11*, whereas a truncated *MAT1-1-11* and two complete genes *MAT1-1-1* and *MAT1-1-10* were present in the *MAT1-1* idiomorph. The third *Mes-20* type of gene arrangement was observed in four yellow morel species, wherein two genes, *MAT1-1-1* and *MAT1-1-10,* were located on the *MAT1-1* idiomorph, while the *MAT1-2* idiomorph carried only the *MAT1-2-1* gene. For *M. rufobrunnea*, the mating genes *MAT1-1-10*, *MAT1-1-1*, and *MAT1-2-1* were arranged in a tandem array and the intergenic length between the *MAT1-1-1* and *MAT1-2-1* was about 7 kb. Both mating types were linked in the mating locus, a form that was regarded as homothallism.

The size of the mating idiomorph, defined as the region of no detectable homology between opposite mating types, was identified by alignment analyses of two amplicon sequences from the same species, with the identical flanking sequences removed. Interspecific variations in the fragment size ranged from 7.2 to 11.8 kb in the *MAT1-1* idiomorphs and from 6.4 to 15.9 kb in the *MAT1-2* idiomorphs (Table 2). However, they were similar within the groups regarding the above three mating idiomorphs. For three species from the *M. importuna* group, the lengths of the *MAT1-1* idiomorphs varied from 10.2 to 11.5 kb, which were larger than the 6.6 to 7.1 kb of the *MAT1-2* idiomorph. The lengths of the *MAT1-1* idiomorph of *M. purpurascena* and *M. pulchella* were 9.4 and 11.8 kb, respectively, which were shorter than their *MAT1-2* idiomorphs (15.9 kb). For three yellow morel species from the *Mes-20* group, the lengths of the *MAT1-1* and *MAT1-2* idiomorphs were similar and ranged from 6.4 to 8.0 kb. Regarding another yellow morel, *Mes-15*, the 5′ and 3′ flanking genes were incomplete, and the size of the *MAT1-1* and *MAT1-2* idiomorphs could not be verified. For the homothallic *M. rufobrunnea*, the size of the *MAT* locus, defined as the distance between the stop codon of the 5′ flanking gene (*SDH2*) and the start codon of the 3′ flanking gene (*MBA1*), was 16.7 kb in length.

Generally, non-coding regions of the *MAT* loci showed more inter-specific variability than the coding portions [20]. Between the groups of the above three mating idiomorph types, homology was mainly present in the mating genes and flanking genes, while high sequence divergence was observed in the non-coding regions (Figure 2). This was the same in both *M. importuna* and *M. purpurascena* groups, although they were black morels. However, between intra-group species, a high sequence homology in both coding and non-coding regions was observed, which could be between *M. sextelata, M. eximia*, and *M. importuna*, as well as between the three yellow morels.

Except for strain M115 of *Mes-15*, the 5′ and 3′ flanking sequences of the *MAT1-1* and *MAT1-2* idiomorphs harbored the same genes, i.e., *SDH2* and *MBA1*, which were highly conserved across the nine other species. The BLASTn searchers against the M115 genome were run using the *SDH2* and *MBA1* genes as the query. These searches indicated that *SDH2* was only in the scaffold carrying the *MAT1-1* idiomorph, while *MBA1* was in the scaffold harboring the *MAT1-2* idiomorph.

The orientation of the mating genes was well-conserved in these morel species. When the mating-type locus was oriented toward the direction of the *MAT1-2-1* gene, the *MAT1-1-1* gene was always arranged in the same direction. However, the *MAT1-1-10* and *MAT1-1-11* genes, as well as the flanking genes *SDH2* and *MBA1*, were oriented in the opposite direction as the *MAT1-2-1* gene.

### 3.3. Unconventional Integration of The MAT Loci in M. purpurascena and M. pulchella

The alignment analyses in *M. purpurascena* and *M. pulchella* revealed a complete *MAT1-1-11* gene of 1838 and 1828 bp in length, respectively, located in the *MAT1-2* idiomorph between the *SDH2* and *MAT1-2-1* gene. On the other hand, the *MAT1-1* idiomorph carried a truncated *MAT1-1-11* fragment of a length of around 900 bp. A comparison of the sequences with the *MAT1-1-11* of the *MAT1-2* idiomorph from the same species indicated that about 900 bp were missing at the 5′ end of the gene in the *MAT1-1* idiomorph, although there was 97% sequence similarity between the complete gene and the truncated sequences. Analyses of synteny revealed several homologous regions in the flanking regions of the *MAT1-1-11* between the *MAT1-1* and *MAT1-2* idiomorphs (Figure 3a), unlike that in the *M. importuna* group, where there was a lack of similar sequences between the opposite-mating idiomorphs (Figure 3b). We hypothesized that unconventional integration occurred at the *MAT* loci between the opposite-mating type idiomorphs of *M. purpurascena* and *M. pulchella*. The complete *MAT1-1-11* in the *MAT1-2* idiomorphs were probably acquired from the *MAT1-1* idiomorph, partial fragments were retained in the *MAT1-1* idiomorph, and the sizes of the *MAT1-2* idiomorphs of these two species were increased to 15.9 kb. Similar events were also reported in other ascomycetes [21,22]. In seven species in the family Ophiostomatales, an ancient inversion event resulted in the integration of the ancestral *MAT1-1-1* gene into the *MAT1-2* idiomorph and survival as the truncated *MAT1-1-1* gene [21]. For the apple cancer pathogen *Valsa mali*, two flanking genes, *COX13* and *APN2*, were co-opted into the *MAT* locus and unconventional recombination occurred between *MAT1-1* and *MAT1-2* idiomorphs, resulting in reverse insertion in the *MAT1-2* idiomorph [22].

Phylogenetic analyses indicated that *M. sextelata* (*Mel-6*), *M. importuna* (*Mel-10*), and *M. exuberans* (*Mel-9*) represented three species lineages of an early evolutionary origin in China, whereas the species lineages of another 27, including *M. purpurascena* (*Mel-20*) and *M. pulchella* (*Mel-31*), diversified between the middle-Miocene and the present [12]. Therefore, the structure of the mating idiomorphs of the *M. importuna* type should be of earlier origin in the Elata clade species, and unconventional integration of the *MAT* locus was observed in *M. purpurascena* and *M. pulchella* due to their evolution from species lineages in the Elata clade. However, to confirm this hypothesis, more information on the *MAT* loci from additional morel species will be necessary.

Some small fragments with sequences that were homologous to the *MAT1-1-10* gene were observed in the regions between the *SDH2* and *MAT1-2-1* in the *MAT1-2* idiomorph of four yellow morel species (Figure 3c). The sizes and positions of homologous fragments differed according to the species, while the sequence identities of the *MAT1-1-10* gene varied from 60% to 80%. Despite their homology to the *MAT1-1-10* gene, these sequences neither coded for any proteins nor contained ORFs, as a start codon was absent.

### 3.4. Transcription Analyses of The Mating Genes

RT-PCR analyses revealed that four mating genes, *MAT1-1-1*, *MAT1-1-10*, *MAT1-1-11*, and *MAT1-2-1,* were constitutively expressed in the tested strains incubated on PDA for 10 days. The *MAT1-1-11* gene could be successfully amplified with cDNA from the *MAT1-1* mating-type strains of *M. sextelata*, *M. eximia*, and *M. importuna*, as well as cDNA from the *MAT1-2* mating type strains of *M. purpurascena* and *M. pulchella*. However, the truncated *MAT1-1-11* fragment could not be detected in the *MAT1-1* mating type strains of *M. purpurascena* and *M. pulchella*, despite designing more primer pairs to amplify it (Appendix A).

The coding regions of the mating genes were verified by comparing the RT-PCR sequences with the corresponding DNA sequences, and the encoded protein sequences were obtained (Table 3). For these ten morel species, the *MAT1-1-1* gene varied in length from 1645 to 1766 bp in lengths, which had two introns and encoded proteins of 509 to 547 amino acids, respectively. The sizes of the *MAT1-2-1* gene varied from 1197 to 1343 bp, while the ORFs carried three introns and the coded proteins contained 329 to 384 amino acids. Both mating genes were longer in four yellow morels and *M. rufobrunnea* than in the five black morel species, which was also the case with the numbers of encoded amino acids. The *MAT1-1-11* gene, harboring five introns, had similar lengths and numbers of encoded amino acids in the five black morel species.

### 3.5. Alternative Splicing of The MAT1-1-10 Gene

The validation of the coding regions of the *MAT1-1-10* gene revealed some interesting phenomena, wherein this gene underwent alternative splicing (AS) to produce different splice variants, and the types of AS varied between species. Among all AS events in the *MAT1-1-10* gene, there were two main types, intron retention and alternative 5′ or 3′ splice sites (Figure 4b,c), with the regular splicing mode (Figure 4a). Intron retention is the most common type of AS in fungi [23]. In this study, except for the third intron, the other four introns in the *MAT1-1-10* gene were retained (Figure 4b), and the retention of the fifth intron (B4) was the most frequent type of AS in the *MAT1-1-10* gene. In some species, for example, in *M. sextelata*, combinations of four types of intron retention were presented (Table 4), which revealed that the second, fourth, and fifth introns were retained alone or together, while the retention of the second intron resulted in premature termination of translation, so the kinds of combination including this retention could silence the gene.

Another type of AS in the *MAT1-1-10* gene was the presence of alternative 5′ or 3′ splice sites, which were present in all except the fourth intron (Figure 4c), and all the splice sites were canonical 5′ GU.....3′AG. In the second intron, alternative 5′ and 3′ splicing was observed whereas, in the first, third, and fifth introns, only one type occurred. In *Mes-6*, there were two alternative 5′ splice sites in the second intron, and both RNA splice products differed by three amino acids. In some species, alternative 5′ or 3′ splice sites also induced gene silencing (Table 4). Besides, in *M. sextelata* and *M. rufobrunnea*, alternative 3′ splice sites were observed in the third intron of the *MAT1-2-1* gene, and both RNA splice products resulted in the premature termination of translation.

To date, the *MAT1-1-10* gene was only observed in the *Morchella* species, and its biological function was unknown. During vegetative growth, this gene could normally express and undergo AS to produce different mRNAs that encoded proteins of different sizes. We hypothesize that this mating gene could play an important role during the asexual phase.

## 4. Discussion

### 4.1. Mating Systems in Morchella

The characteristics of the mating idiomorphs from five black morel species, *M. sextelata*, *M. eximia*, *M. importuna*, *M. purpurascena*, and *M. pulchella*, and three yellow morel species, *Mes-6*, *Mes-19*, and *Mes-20*, indicated heterothallism. The single-ascospore isolates carried either the *MAT1-1* or the *MAT1-2* idiomorph. Furthermore, the two idiomorphs had the conserved flanking sequences (*SDH2* and *MAB1*), which confirmed that they occupied the same chromosomal locus. Nevertheless, in *M. rufobrunnea* from the Rufobrunnea clade, the mating genes *MAT1-1-10*, *MAT1-1-1*, and *MAT1-2-1* were linked and contained in the haploid genome, while the mating locus also had the conserved flanking genes *SDH2* and *MAB1*, which indicated that *M. rufobrunnea* was homothallic.

For *Mes-15*, the mating genes *MAT1-1-10*, *MAT1-1-1*, and *MAT1-2-1* were contained in the single-ascospore strain M115 genome. However, the *MAT1-1-10* and *MAT1-1-1* genes were in scaffold 677 that carried four genes, viz., *CPSF6*, *TFA1*, *ATP4*, and *SDH2* in the 5′ flanking regions like the structures observed in *M. importuna* and *Mes-20* [16,17], except for the absence of *MBA1* from the 3′ flanking sequences. Similarly, the *MAT1-2-1* and two 3′-flanking genes, *MBA1* and *SLA2,* were located in another scaffold 700, although *SDH2* was not found in the 5′ flanking regions. We could not determine whether *Mes-15* was a homothallic or pseudohomothallic species, however, the genome data from a yellow morel species, *M. peruviana* (JGI Project Id 1239452), gave some clues. In this genome, the mating genes *MAT1-1-10*, *MAT1-1-1*, and *MAT1-2-1* were present in scaffold 12, while the adjacent genes *SDH2* and *MBA1* were located in the 5′ and 3′ flanking regions, respectively, but the distance between the *MAT1-1-1* and *MAT1-2-1* was over 38 kb. These features indicated that *M. peruviana* was homothallic. Analysis of synteny indicated that the scaffold 677 of strain M115 was highly homologous with the *MAT1-1-1* region of the scaffold 12 of *M. peruviana*, while the scaffold 700 shared identity with the *MAT1-2-1* region (Appendix A). Based on these findings, we hypothesized that *Mes-15* was a homothallic species, and the *MAT1-1-1* and *MAT1-2-1* genes were not tightly linked, like in *M. peruviana.*

By analyzing the mating-type ratios of the single-spore populations, Chai et al. [17] considered three black morel species, *M. sextelata*, *M. importuna*, and *Mel-20*, to show heterothallism. Du and Yang [16] reported that 15 black morel species and 19 yellow morel species were heterothallic, and among the 38 morel species studied, the *MAT1-1-10* gene could be successfully amplified in 34 species except for four black morel species, whereas the *MAT1-1-11* gene was found in only seven black morel species. In contrast, recent studies have reported that *M. importuna* was a pseudohomothallic species [24]. The single-ascospore strain of *M. importuna* could fructify normally [24,25,26,27]. Furthermore, there was a great deviation in the proportion of two opposite mating-type nuclei in the single-ascospore isolates, and it was sometimes difficult to detect the mating-type with less distribution by PCR amplification. Therefore, for other *Morchella* species, more detailed research may be required to confirm heterothallism or pseudohomothallism.

Whether homothallism or heterothallism is the ancestral reproductive state in ascomycetes is still a controversial issue [28,29]. In *Morchella*, the Rufobrunnea clade consisting of *M. rufobrunnea* and *M. Anatolica*, demonstrate the earliest diverging branch, followed by the origin of the Esculenta and Elata sister clades from approximately 30 million years later [11,12]. Therefore, the presentation of homothallism in *M. rufobrunnea* should be the ancestral reproductive state in *Morchella*. During evolution, some species have retained homothallic modes, whereas others have turned to heterothallism, as heterothallism would offer the advantages of enhanced genetic diversity and adaption to the environment [30]. The *MAT1-1-10* gene was the primary mating gene in *Morchella*, like the *MAT1-1-1* and *MAT1-2-1* genes, and the *MAT1-1-11* gene was integrated into the mating loci during the process of evolution. However, further research is needed to confirm whether this gene was distributed in all species or only in some species from the Elata clade.

### 4.2. Unconventional Integration of the MAT Locus in Some Morel Species

Generally, the mating-type loci in fungi have often evolved extensive suppression of recombination [31]. However, unconventional recombination or crossover at the *MAT* locus between the opposite mating types has also been demonstrated in several ascomycetes [21,22]. In Botryosphaeriaceae, some heterothallic species had *MAT1-2-**5* fragments in the *MAT1-1* idiomorph. Furthermore, they showed a truncated *MAT1-1-**8* gene in the *MAT1-2* idiomorph [28]. In some *Phyllosticta* species, a truncated version of the *MAT1-1-8* gene was in the *MAT1-2* strains [32]. It has been proposed that these partial gene fragments in the mating-type locus of heterothallic strains can arise either from their heterothallic ancestors that underwent unequal crossing over/recombination events or from their homothallic ancestors that experienced two independent deletion events [28,32].

We hypothesized that unconventional integration occurred at the *MAT* locus of *M. purpurascena* and *M. pulchella*. Phylogenetic analyses indicated that the divergence time of *M. sextelata* and *M. importuna* was earlier than that of *M. purpurascena* and *M. pulchella* [12]. Therefore, the structure of mating idiomorphs of the *M. importuna* group-the *MAT1-1-1*, *MAT1-1-10*, and *MAT1-1-11* in the *MAT1-1* idiomorph and only the *MAT1-2-1* in the *MAT1-2* idiomorph-should be of a relatively older origin in the species from the Elata clade. During evolution, unclear integration events occurred between the opposite idiomorphs in *M. purpurascena* and *M. pulchella*, the *MAT1-1-11* gene was restructured into the *MAT1-2* idiomorph, and the partial fragment was retained in the *MAT1-1* idiomorph, giving rise to the mating idiomorph of the *M. purpurascena* type.

Based on the analyses of the genome data of some morel species submitted to JGI, we observed that the mating idiomorph of the *M. importuna* type in *M. semilibera* (*Mel-3*), *M. punctipes* (*Mel-4*), and *M. disparilis*, but some other species including *M. snyderi* (*Mel-12*), *M. angusticeps* (*Mel-15*), *M. hispaniolensis* (*Mel-18*), *M. eohespera* (*Mel-19*), and *M. brunnea* (*Mel-22*), had the mating idiomorph of the *M. purpurascena* type. Therefore, the integration events had not happened by accident in some species but had occurred through evolution from species lineages.

### 4.3. Alternative Splicing of The MAT1-1-10 Gene

Alternative splicing is a regulated process in eukaryotes, which leads to expansion in the form and function of the proteome [33]. It occurs in 95–100% of human genes [33], 60% of plant genes [34], 8.6% of Basidiomycota, and 6.0% of Ascomycota genes [35]. However, recent studies indicate that AS occurs in fungi at a much higher frequency than reported earlier [36]. For example, rates of AS in *Trichoderma longibrachiatum* and *Cryptococcus neoformans* were 48.9% [37] and 59% [38], respectively. AS serves several regulatory functions in fungi, including environmental adaptation, protein localization, meiosis, and gene expression [36]. Genes with regulatory functions, such as transcription factors or signal transducers exhibit high levels of alternative splicing, while basic enzymatic functions show less splice variation [39,40].

The *MAT1-1-10* was an ancestral mating gene in *Morchella*, which was reported in recent years [7], and little was known about its functions. In this study, the alternative splicing phenomenon of this gene was observed in 10 morel species from three clades, leading to different mRNAs that encoded proteins with different amino acid lengths. AS occurred in the mycelia that were incubated in normal growing conditions and were not subjected to environmental pressure, and it is unknown whether these splice variants would exhibit different biological functions during vegetative growth, which has also aroused our curiosity about the roles of the *MAT1-1-10* gene.

## 5. Conclusions

The results from our study showed that homothallism should be the ancestral reproductive state in *Morchella.* The mating-type loci from ten morel species have been characterized and compared. It confirmed that *M. rufobrunnea*, a representative species of the earliest diverging branch of true morels, along with another yellow morel *Mes-15*, were homothallic, and another eight morel species were heterothallic. Three types of composition were observed in the mating idiomorphs of eight heterothallic species, and an unconventional integration event occurred in two black morel species and resulted in the importation of the *MAT1-1-11* gene into the *MAT1-2* idiomorph and survival as the truncated gene. The primer pairs p11-1f/p11-2r could successfully amplify the mating-type idiomorphs of heterothallic species from the Elata and Esculenta clades, which make it possible to obtain more information on the *MAT* loci from additional morel species by PCR.

## Figures and Tables

**Figure 1 jof-08-00746-f001:**
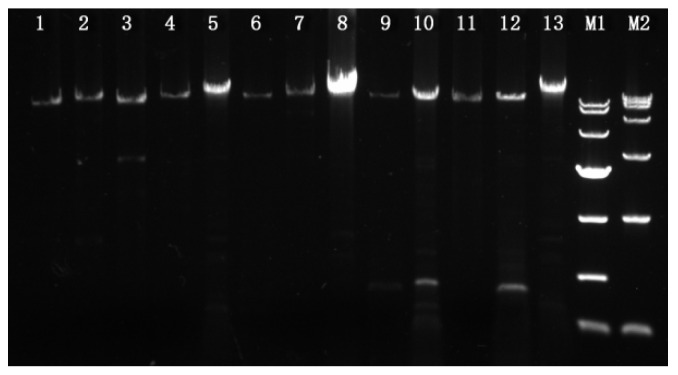
Agarose gel electrophoresis of PCR products of the mating idiomorphs. Lane 1, 3, 5, 7, 9, 11: PCR products from the *MAT1-2* type strains YAASMQM-4, YAASMHL47, YAASM23-22, YAASM131-12, YAASM50-09, and YAASM43-08; Lane 2, 4, 6, 8, 10, 12: PCR products from the *MAT1-1* type strains YAASMQM-23, YAASMHL1, YAASM23-11, YAASM131-9, YAASM50-38, and YAASM43-24; Lane 13: PCR products from strain YAASMVR; M1: DL10000 DNA Marker (10,000, 7000, 4000, 2000, 1000, 500, 250); M2: DL15000 DNA Marker (15,000, 10,000, 7500, 5000, 2500, 1000, 250).

**Figure 2 jof-08-00746-f002:**
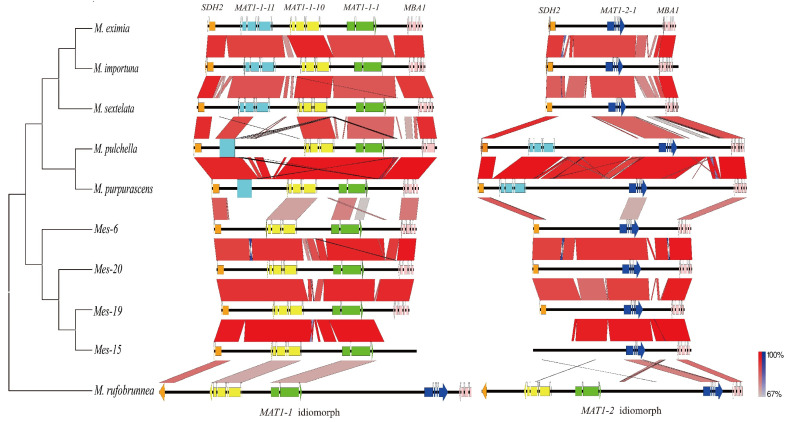
Pairwise comparison of the mating idiomorphs between species of *Morchella*. The black horizontal lines represent genomic sequences, and the color-coded arrows represent coding sequences in the forward (right-oriented arrow) or reverse (left-oriented arrow) strand. The red or blue boxes between genomic sequences indicate pairwise similarity based on BLASTn; red boxes indicate that both regions are in the same orientation, while blue boxes indicate that the regions are oriented in the opposite directions. Neighbor-joining (NJ) phylogenetic analysis between *Morchella* species based on a multi-locus dataset (ITS, *RPB1*, *RPB2*, LSU, and *EF1**-**a*).

**Figure 3 jof-08-00746-f003:**
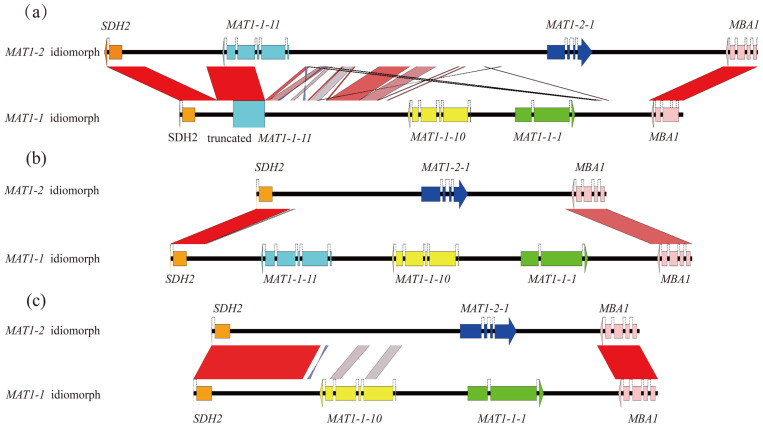
Comparison of the opposite-mating idiomorphs from the same species. Comparison of the opposite-mating idiomorphs of (**a**) *M. pulchella*, (**b**) *M. eximia*, (**c**) *Mes-19*. Black horizontal lines represent genomic sequences, and color-coded arrows represent coding sequences. Red or blue boxes between genomic sequences indicate pairwise similarity based on BLASTN. Red boxes indicate that both regions are in the same orientation, and blue boxes indicate that the regions are oriented in opposite directions.

**Figure 4 jof-08-00746-f004:**
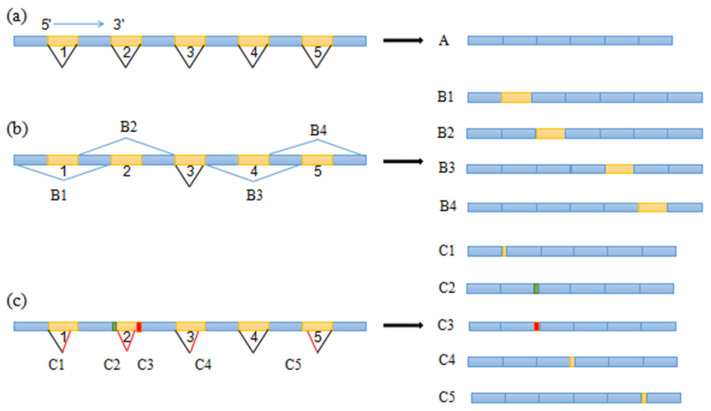
A schematic diagram showing the different types of AS events in the *MAT1-1-10* gene. (**a**) Normal splicing. (**b**) Intron retention: B1, Retention of the first intron; B2, Retention of the second intron; B3, Retention of the fourth intron; B4, Retention of the fifth intron. (**c**) Alternative 5′ and 3′ splicing: C1, Alternative 3′ splicing in the first intron; C2, Alternative 5′ splicing in the second intron; C3, Alternative 3′ splicing in the second intron; C4, Alternative 3′ splicing in the third intron; C5, Alternative 5′ splicing in the fifth intron.

**Table 1 jof-08-00746-t001:** The morel strains used in this study.

Strain	Species	Code	Ascocarp	Origin	Mating-Type Idiomorph
YAASMHL1	*M. sextelata*	*Mel-6*	YAASMHL	Sichuan, China	*MAT1-1*
YAASMHL47					*MAT1-2*
YAASMQM-23	*M. eximia*	*Mel-7*	YAASMQM	Tibet, China	*MAT1-1*
YAASMQM-4					*MAT1-2*
YAASM23-11	*M. purpurascena*	*Mel-20*	YAASM23	Yunnan, China	*MAT1-1*
YAASM23-22					*MAT1-2*
YAASM131-9	*M. pulchella*	*Mel-31*	YAASM131	Yunnan, China	*MAT1-1*
YAASM131-12					*MAT1-2*
YAASM50-38		*Mes-6*	YAASM50	Shanxi, China	*MAT1-1*
YAASM50-09					*MAT1-2*
M115		*Mes-15*	YAASM2689	Yunnan, China	*MAT1-1, MAT1-2*
YAASM43-24		*Mes-19*	YAASM43	Sichuan, China	*MAT1-1*
YAASM43-08					*MAT1-2*
YAASMVR	*M. rufobrunnea*			American	*MAT1-1, MAT1-2*

**Table 2 jof-08-00746-t002:** Organization of the mating idiomorphs.

Species	Code	*MAT1-1* Idiomorph			*MAT1-2* Idiomorph		
		Gene Arrangements	Lengths (kb)	GenBank Accession	Gene Arrangements	Lengths (kb)	GenBank Accession
*M. sextelata*	*Mel-6*	*MAT1-1-1, MAT1-1-10, MAT1-1-11*	11.5	MN589921	*MAT1-2-1*	7.1	MN589922
*M. eximia*	*Mel-7*	*MAT1-1-1, MAT1-1-10, MAT1-1-11*	10.2	ON622485	*MAT1-2-1*	6.6	ON622484
*M. importuna*	*Mel-10*	*MAT1-1-1, MAT1-1-10, MAT1-1-11*	10.5	KY782630	*MAT1-2-1*	6.7	KY782629
*M. purpurascena*	*Mel-20*	*MAT1-1-1, MAT1-1-10,* truncated *MAT1-1-11*	9.4	MN589929	*MAT1-2-1, MAT1-1-11*	15.9	MN589930
*M. pulchella*	*Mel-31*	*MAT1-1-1, MAT1-1-10,* truncated *MAT1-1-11*	11.8	ON622483	*MAT1-2-1, MAT1-1-11*	15.9	ON622482
	*Mes-6*	*MAT1-1-1, MAT1-1-10*	8.0	MN589927	*MAT1-2-1*	7.3	MN589928
	*Mes-15*	*MAT1-1-1, MAT1-1-10*	-	KY782631	*MAT1-2-1*	-	KY782632
	*Mes-19*	*MAT1-1-1, MAT1-1-10*	7.2	MN589925	*MAT1-2-1*	6.4	MN589926
	*Mes-20*	*MAT1-1-1, MAT1-1-10*	7.8	MN589923	*MAT1-2-1*	7.5	MN589924
*M. rufobrunnea*		*MAT1-1-1, MAT1-1-10, MAT1-2-1*	16.7	MN589931			

**Table 3 jof-08-00746-t003:** The variance of the mating genes in ten morel species.

Species	Code	*MAT1-1-1*			*MAT1-2-1*			*MAT1-1-11*		
		DNA (bp)	Intron Numbers	Amino Acids (aa)	DNA (bp)	Intron Numbers	Amino Acids (aa)	DNA (bp)	Intron Numbers	Amino Acids (aa)
*M. sextelata*	*Mel-6*	1730	2	537	1199	3	329	1823	5	511
*M. eximia*	*Mel-7*	1676	2	519	1203	3	330	1825	5	511
*M. importuna*	*Mel-10*	1694	2	525	1197	3	329	1825	5	512
*M. purpurascena*	*Mel-20*	1651	2	511	1247	3	344	1838	5	511
*M. pulchella*	*Mel-31*	1645	2	509	1247	3	344	1828	5	511
	*Mes-6*	1757	2	547	1328	3	385			
	*Mes-15*	1766	2	543	1304	3	377			
	*Mes-19*	1751	2	545	1301	3	376			
	*Mes-20*	1751	2	545	1286	3	371			
*M. rufobrunnea*		1760	2	546	1343	3	384			

**Table 4 jof-08-00746-t004:** The types of alternative splicing in the *MAT1-1-10* gene.

Species	Code	Total Numbers of Clones	Types of Splicing ^a^	Numbers of Clones	Intron Numbers	DNA (bp)	Amino Acids
*M. sextelata*	*Mel-6*	21	A	11	5	1723	482
			B4	3	4	1689	490
			B3 + B4	3	3	1689	510
			B2 + B4	1	3	- ^b^	-
			B2 + B3 + B4	3	2	-	-
*M. eximia*	*Mel-7*	10	B4	10	4	1661	479
*M. importuna*	*Mel-10*	10	B4	10	4	1688	488
*M. purpurascena*	*Mel-20*	10	A	7	5	1737	482
			C3	3	5	1737	486
*M. pulchella*	*Mel-31*	10	C3	6	5	1737	478
			C4	2	5	1737	482
			C3 + C4	2	5	1737	486
	*Mes-6*	10	A	2	5	1762	493
			C1	1	5	-	-
			C1 + B4	1	4	-	-
			B4	2	4	1654	478
			C5	3	5	1704	473
			C5-2 ^c^	1	5	1733	476
	*Mes-15*	10	A	8	5	1760	491
			C5	1	5	-	-
			C1 + C5	1	5	-	-
	*Mes-19*	14	A	11	5	1760	491
			B4	3	4	1760	512
	*Mes-20*	10	A	1	5	1757	491
			B4	6	4	1648	476
			C5	1	5	1699	420
			B1 + C5	1	4	1699	489
			B1 + B4 + C2	1	3	1649	474
*M. rufobrunnea*	*Mruf*	11	B4	4	4	1698	485
			C3	3	5	1850	515
			C3 + B4	3	4	1698	489
			B1 + C3	1	4	-	-

^a^: Represented by the codes in Figure 4; ^b^: Gene silencing; ^c^: Another alternative 5′ splice site in the fifth intron.

## Data Availability

The *MAT1-1* idiomorph sequences of *M. sextelata**, M. eximia*, *M. importuna**, M. purpurascena*, *M. pulchella*, *Mes-6*, *Mes-15*, *Mes-19*, and *Mes-20* can be found at GenBank accession numbers MN589921, ON622485, KY782630, MN589929, ON622483, MN589927, KY782631, MN589925, MN589923. The *MAT1-2* idiomorph sequences of each species can be found at GenBank accession numbers MN589922, ON622484, KY782629, MN589930, ON622482, MN589928, KY782632, MN589926, MN589924. The *MAT* locus sequences of *M. rufobrunnea* can be found at GenBank accession number MN589931. The genome data of *Mes-15* was under accession number GWHBJCX00000000 in the Genome Warehouse in the National Genomics Data Center.

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
