# Peer review of "Organization and Unconventional Integration of the Mating-Type Loci in Morchella Species"

_jof, 2022, doi:10.3390/jof8070746_

Round 1

Reviewer 1 Report

The manuscript ” Organization and Unequal Recombination of The Mating-type 2 Loci in Morchella Species”by Chai et al.  reports the mating-type genes of 10 morel species. The work is continuation of the research that the group has already before published. On the bases of DNA isolated from a mycelium grown from  single ascospores from each species and by PCR primers for mating-type genes it was  concluded that several black morels are heterothallic. On the bases of this previous work the genes at the 5´- and 3´of  Mat1-1 and Mar1-2 idiomorphs were defined and by long-rage PCR the genomic sequences of the idiomorphs of the ten morel species were analyzed , including also the black morels investigated earlier. The present work includes also four species that belong to the yellow group and one to “blushing morels” M. rufobrunnea. MAT1-1-1 (alfa-box gene) was found in all studied species as well MAT1-2-1 (HMG gene). In heterothallic species they were identified in DNA from mycelium grown from different ascospores. In addition MAT1-1 idiomorph (I like better locus) contains the genes MAT1-1-11 and MAT1-1-10 that had previously been identified in black morels. New in this work is that in two black morel species (M. pulchella and M. purpurascens) unequal recombination had taken place betweenMat1-1-1 and MAT1-2-1 idomorphs transferring Mat1-1-11 gene to Mat1-2-1 idiomorph and leaving traces of the gene in the MAT1-1 idiomorph. New result is also the identification of the high number of alternative splicing of MAT1-1-10 gene. In addition two species Mes 15 (yellow morel) and M.rufobrunnea are shown to be homothallic , the mycelium from sinle ascospores contain both idiomorphs, and the black morel M.importuna pseudohothallic, although not shown experimentally in the present manuscript but in ref. 13. The result are presented clearly by Figs and Tables, the text is well written and the data is interesting. The manuscript is recommended to be published in Journal of Fungi.

Some small comments are possible to make and should be corrected.

Abstract

“Mes-15, were confirmed to be homothallism”, to be homothallic

Introduction

“morel species have been reported [14, 15]” should include [7, 14, 15]

Results

It is somewhat tedious when the species names, strain names and codes are used in the text, although they are listed in Table 1.

The sentence starting at line 193 and ending at line 197 should be rewritten at least as two sentences

Table 2  MAT1-2 idiomorph for M. purpurascena and M. pulchella should include Mat1-1-11, since MAT11-1 idiomorphs contains the truncated MAT1-1-11

Discussion

Any explanation why alternative splicing is detected only in MAT1-1-10 gene?

 “pseudohomothallic species possessing nuclei of opposite mating types  in multinuclear isolates are all capable of self-fertility and outcrossing [9].” In reference 23 it is suggested that M.importuna is a pseudohomothallic fungus. In the present manuscript only heterothallic. Unfortunately it was not possible to read reference23. Obviously the ascospores of M. importuna contain at least two nuclei and the number of ascospores per ascus is less????   

Are the genes MAT1-1-10 and MAT1-1-11 specific for morels and what their function could be. In the text it said that MAT1-1-10 could be necessary for asexual reproduction, but no more explanation.    

Author Response

  1. Abstract

“Mes-15, were confirmed to be homothallism”, to be homothallic

Response: Thanks. We have changed it.

  1. Introduction

“morel species have been reported [14, 15]” should include [7, 14, 15]

Response: Thanks. This section has been modified, and the corresponding references have changed.

  1. Results

(1) It is somewhat tedious when the species names, strain names and codes are used in the text, although they are listed in Table 1.

Response: Thanks. The phylospecies that have not been described are only represented by phylospecies codes, for example four yellow morel species in this manuscript. Although the codes of five black morel species were also listed in Table 1, we used the species names in the text. There were a MAT1-1 and a MAT1-2 in the strain for each species, and we had to use the strain names to state in some cases.   

(2)The sentence starting at line 193 and ending at line 197 should be rewritten at least as two Sentences

Response: Thanks. We have rewritten this section to two sentences.

(3)Table 2 MAT1-2 idiomorph for M. purpurascena and M. pulchella should include Mat1-1-11, since MAT11-1 idiomorphs contains the truncated MAT1-1-11

Response: Thanks. For M. purpurascena and M. pulchella, the MAT1-1-11 gene in the MAT1-2 idiomorph,and the texts had been hidden due to improper column width in Table 2 were abnomal, and we have modified it.

  1. Discussion

(1)Any explanation why alternative splicing is detected only in MAT1-1-10 gene?

Response: Thanks. Alternative splicing was also detected in the MAT1-2-1 gene in M. sextelata and M. rufobrunnea, and we had illustrated them in the Results 3.5. We cannot explain why alternative splicing is detected only in the MAT1-1-10 gene now, for the biological functions of this mating gene in Morchella is unknown. The biological functions of MAT1-1-10 gene, including the diversities of alternative splicing, will be studied in our future research.

(2) “pseudohomothallic species possessing nuclei of opposite mating types in multinuclear isolates are all capable of self-fertility and outcrossing [9].” In reference 23 it is suggested that M.importuna is a pseudohomothallic fungus. In the present manuscript only heterothallic. Unfortunately it was not possible to read reference23. Obviously the ascospores of M. importuna contain at least two nuclei and the number of ascospores per ascus is less????

Response: Thanks. The numbers of ascospores per ascus are still 8. In the present manuscript, we verified heterothallism and homothallism by analyzing organization of the MAT loci. In reference 23, single-ascospore populations and 8 single-ascospore strains in an ascus were analyzed. The abstract of the reference 23 is as follows:

Abstract: The mating-types of F1 ascocarps, single-ascospore populations and 8 single ascospore strains in an ascus were analyzed based on the crossing of single-ascospore strains YPL6-1 and YPL6-3 of Morchella importuna which harbored MAT1-2 and MAT1-1 idiomorph respectively in their genome-sequencing data. Under the conditions of sowing separately and mix-sowing, strains YPL6-1 and YPL6-3 could fructify normally, and the distribution of mating type in the stipe was related to the parent strain. When the mating type tests were carried out by PCR amplification in 235 single-ascospore strains, something interesting happened: the electrophoretic bands of MAT1-1-1 gene in some strains were bright, but the bands of MAT1-2-1 were weak. In other strains, the MAT1-1-1 band was weak, while the MAT1-2-1 band was bright. Meanwhile, there were strains that two mating gene bands were bright, or strains that one mating gene band was bright while the opposite mating gene band unappeared. Ten asci were separated from three ascocarps and corresponding single ascospores were isolated from each ascus, and the mating types of these single-ascospore strains were analyzed. As a result, the same phenomenon occurred. The single-ascospore strains showing bright MAT1-1-1 band and weak or no MAT1-2-1 band originated from no more than 4 spores in an ascus, and vice versa. The PCR amplicons of MAT loci in the YPL6-1 and YPL6-3 were sequenced respectively by nanopore approach, and corresponding experiments were repeated twice. The alignment analysis indicated that there were 99.63% and 99.81% MAT1-2 idiomorphs, and 0.37% and 0.19% MAT1-1 idiomorphs in the strain YPL6-1, meanwhile, the strain YPL6-3 contained 99.45% and 99.74% MAT1-1 idiomorphs, and 0.55% and 0.26% MAT1-2 idiomorphs. The result confirmed that these two single-ascospore strains were actually heterokarytic, however, there was a great deviation in proportion of two mating type nuclei. It is speculated that all ascospores in M. importuna are heterokarytic, and the opposite mating type nuclei are asymmetrically distributed in mycelia germinated from single ascospore. Therefore, M. importuna is a pseudohomothallic.

(3) Are the genes MAT1-1-10 and MAT1-1-11 specific for morels and what their function could be. In the text it said that MAT1-1-10 could be necessary for asexual reproduction, but no more explanation.

Response: Thanks. Although the biological functions the genes MAT1-1-10 and MAT1-1-11 were unknown, as mentioned in the text, alternative splicing of the MAT1-1-10 gene occurred in the mycelia that were incubated in normal growing conditions and were not subjected to environmental pressure, and AS produced different mRNAs that encoded proteins of different sizes. Therefore, we speculated that the MAT1-1-10 could be necessary for asexual reproduction.

Reviewer 2 Report

The manuscript describes the variations of the mating type loci in Morchella species based on sequence analysis. It found that the mating type loci are diversified through several recombination events that generate different mode of sexual reproduction in this species. The main text is well-written and carries scientifically sound findings which makes the paper worthy of publication.

Author Response

Thanks. No comments.

Reviewer 3 Report

This is a very good study.

1. If you add a reference to other mushroom mating-related studies in the introduction, it would be more complete.

2. The clade of each strains were described by citing other studies. So readability is reduced. Could you please indicate it in a table, etc. in the manuscript?

Author Response

1.If you add a reference to other mushroom mating-related studies in the introduction, it would be more complete.

Response: Thanks. We have added the mating-related studies in ascomycetes mushroom Cordyceps and Tuber in the introduction (reference 14 and 15).

2.The clade of each strains were described by citing other studies. So readability is reduced.Could you please indicate it in a table, etc. in the manuscript?

Response: Thanks. Phylogenetic analysis between species in this study had shown in Figure 2, and their branch relationship with other morel species were described by citing other studies. So we think there’s no need to indicate all the origin of the branch in the text.

Reviewer 4 Report

See the minor suggested edits on the pdf returned.  The only place I suggest you may need to verify deals with the 'unequal recombination.  This usually occurs when random repeats can mislaid so a crossover can expand and contract the copy number in progeny.  Are the alleles of the different MAT-1 genes similar enough to allow pairing?  Could there be regions of inverted repeats?  If you know that is not the case it should be mentioned.  If not, a Pustel matrix using the whole region on both axes may reveal regions that could lead toe changes and inversions.  

Author Response

See the minor suggested edits on the pdf returned. The only place I suggest you may need to verify deals with the 'unequal recombination. This usually occurs when random repeats can mislaid so a crossover can expand and contract the copy number in progeny. Are the alleles of the different MAT-1 genes similar enough to allow pairing? Could there be regions of inverted repeats? If you know that is not the case it should be mentioned. If not, a Pustel matrix using the whole region on both axes may reveal regions that could lead toe changes and inversions.

Response: Thanks. We have modified the text according to the suggested edits. We accept that the formulation “unequal recombination” is not proper, because there are lack of similar sequences and regions of inverted repeats between the opposite-mating idiomorphs in the ancestral type (the M. importuna type). We have revised “unequal recombination” to “ unconventional integration” in the manuscript, for its occurrence mechanism is still unclear.